# UV-Activated Au Modified TiO_2_/In_2_O_3_ Hollow Nanospheres for Formaldehyde Detection at Room Temperature

**DOI:** 10.3390/ma16114010

**Published:** 2023-05-26

**Authors:** Su Zhang, Baoyu Huang, Zenghao Jiang, Junfan Qian, Jiawei Cao, Qiuxia Feng, Jianwei Zhang, Xiaogan Li

**Affiliations:** 1School of Microelectronics, Dalian University of Technology, Dalian 116024, China; postsuz@163.com (S.Z.); huangby@dlut.edu.cn (B.H.); mistyayaya@163.com (Z.J.); junfanqian@126.com (J.Q.); c_jw2000@163.com (J.C.); lixg@dlut.edu.cn (X.L.); 2School of Information and Control Engineering, Qingdao University of Technology, Qingdao 266520, China; fengqiuxia@qut.edu.cn; 3School of Artificial Intelligence, Dalian University of Technology, Dalian 116024, China; 4Key Laboratory of Integrated Circuit and Biomedical Electronic System, Dalian University of Technology, Dalian 116023, China

**Keywords:** Au modified TiO_2_/In_2_O_3_ hollow nanospheres, room temperature formaldehyde sensors, UV activation, in situ DRIFTS

## Abstract

Au modified TiO_2_/In_2_O_3_ hollow nanospheres were synthesized by the hydrolysis method using the carbon nanospheres as a sacrificial template. Compared to pure In_2_O_3_, pure TiO_2_, and TiO_2_/In_2_O_3_ based sensors, the Au/TiO_2_/In_2_O_3_ nanosphere-based chemiresistive-type sensor exhibited excellent sensing performances to formaldehyde at room temperature under ultraviolet light (UV-LED) activation. The response of the Au/TiO_2_/In_2_O_3_ nanocomposite-based sensor to 1 ppm formaldehyde was about 5.6, which is higher than that of In_2_O_3_ (1.6), TiO_2_ (2.1), and TiO_2_/In_2_O_3_ (3.8). The response time and recovery time of the Au/TiO_2_/In_2_O_3_ nanocomposite sensor were 18 s and 42 s, respectively. The detectable formaldehyde concentration could go down as low as 60 ppb. In situ diffuse reflectance Fourier transform infrared spectroscopy (DRIFTS) was used to analyze the chemical reactions on the surface of the sensor activated by UV light. The improvement in the sensing properties of the Au/TiO_2_/In_2_O_3_ nanocomposites could be attributed to the nanoheterojunctions and electronic/chemical sensitization of the Au nanoparticles.

## 1. Introduction

Formaldehyde (HCHO) is one of the main air pollutants, and when the concentration of formaldehyde exceeds the standard, it can cause a variety of diseases and has a serious impact on the body organs, mainly manifested in strong stimulation, sensitization, and mutagenesis [1,2,3,4]. In serious cases, it can induce cancer. In view of this, the World Health Organization recommends that long-term exposure to formaldehyde should not exceed 0.8 ppb. It is necessary to develop a fast response sensor for formaldehyde [5,6,7,8]. Compared with traditional detection methods, the use of metal oxide semiconductor (MOS)-based sensors to detect formaldehyde has the advantages of high sensitivity, good stability, and low price, and is a research hotspot in the field of gas sensors.

However, most MOS-based sensors usually need to operate at high temperatures (200–450 °C), which is not conducive to the integration of the device, and the sensing material will be destroyed at high temperature, which will affect the service life of the sensor [9,10,11,12,13]. Due to the limitations of thermal excitation, the activation of these sensors by ultraviolet (UV) irradiation is a promising strategy, especially for the metal oxides with good photocatalytic properties such as TiO_2_, ZnO, WO_3_, In_2_O_3_, and SnO_2,_ etc. Under UV activation, the sensing oxides absorb photons, and electrons in the oxides transition from low to high energy levels to generate electron–hole pairs [14,15,16]. The photogenerated electrons and holes increase the concentration of the charge carriers, thus reducing the operating temperature of the sensor [17,18,19]. Previous studies have found that TiO_2_ with a hollow microstructure showed good sensing performance to formaldehyde gas under UV activation. Titanium dioxide (TiO_2_) is a widely studied gas sensing material with n-type response, and the bandgap of TiO_2_ is about 3.2 eV [20,21]. TiO_2_ is widely used in optoelectronics, photocatalysis, gas detection, and other fields [22]. However, the sensitivity and response/recovery performance of pure TiO_2_ still need to be improved, and the baseline resistance of a TiO_2_-based chemiresistive-type sensor at room temperature is about tens of million of ohms, which is too high to be conveniently manufactured. A second phase with very small resistance needs to be introduced to reduce its baseline resistance. Introducing the second phase into a single phase sensing material is an important way to improve the gas sensing performance of sensors at present [23,24,25]. The formation of nanoheterojunctions play an important role in regulating the concentration of the carrier and the height of the barrier in the sensing materials, which is conducive to improving the conductivity of the sensing material and improving the gas sensing performance of the sensors [26,27,28,29]. Indium oxide (In_2_O_3_) is a novel n-type semiconductor, and the indirect bandgap of In_2_O_3_ is about 2.8 eV [30]. Its conductivity is almost the highest among representative semiconductor oxide sensing materials, and the resistance is only a few to tens of thousands of ohms [31,32]. However, there are few reports on In_2_O_3_-based sensors under UV illumination. For example, Wang et al. [33] integrated ultra-thin In_2_O_3_ films into GaInN/GaN UV-LED structures to create a UV-LED sensor. The results showed that the response of the sensor to 726 ppb O_3_ was about 10.2 at room temperature.

As the surface reaction mechanism in the gas sensing response is similar to the catalytic reaction at the gas–solid interface, noble metal modified metal oxides are also often used in gas sensing [34,35]. The introduction of noble metal nanoparticles into the sensing material as a catalyst can accelerate the response speed, improve the sensitivity, enhance the stability, and improve the selectivity. Loading noble metal nanoparticles uniformly on the surface of the sensing materials will promote the gas sensing properties [36,37,38]. The difference between the work function of noble metal and the support metal oxide semiconductor can promote the adsorption and desorption of the sensing material to the gas and accelerate the electron transfer. Kamble et al. [39] improved the performance of the sensor by modifying a WO_3_ thin film with Ag nanoparticles, and the response speed was improved by six times compared with the WO_3_ thin film based sensor. Gu et al. [40] reported a highly selective and sensitive HCHO sensor based on Au/In_2_O_3_ nanocomposites. The sensor had a high response (85.67) to 50 ppm HCHO at low operating temperature (100 °C), and the detection limit could reach 1.42 ppb. Au nanoparticles dispersed on the In_2_O_3_ surface provide more active sites for the selective adsorption reaction and reduce the activation energy, so Au/In_2_O_3_-based sensors have high sensitivity and good selectivity against HCHO at low temperature. The gas sensor based on noble metal modified sensing materials has advantages of good response and low power consumption. Au nanoparticles as catalysts can control the surface chemical reactions through selective adsorption, reduce the activation energy of formaldehyde surface chemical reactions, and make it easier for formaldehyde to react with the chemisorbed oxygen ions on the surface of sensing materials [41]. The decoration of noble metal nanoparticles on the MOSs can improve the reaction between chemisorbed oxygen ions and target gas molecules, finally, enhancing the sensor’s gas sensing properties.

In this work, Au modified TiO_2_/In_2_O_3_ hollow nanospheres were prepared by a simple hydrolysis method at low temperature. First, hollow In_2_O_3_ nanospheres were synthesized by the immersion method using carbon as sacrificial templates. Then, TiO_2_ nanoparticles were deposited on the surface of the obtained In_2_O_3_. Finally, Au nanoparticles were loaded on the TiO_2_/In_2_O_3_ by the chemical reduction method. The microstructure and gas sensing performance of the obtained Au modified TiO_2_/In_2_O_3_ nanocomposites were analyzed, and the results showed that the gas-sensing properties to formaldehyde of the Au/TiO_2_/In_2_O_3_ nanocomposite-based sensor were significantly improved under UV activation at room temperature.

## 2. Materials and Methods

### 2.1. Synthesis of the Au Modified TiO_2_/In_2_O_3_ Nanocomposites

All chemicals were analytical grade reagents and used without further purification. First, the carbon nanospheres used as hard templates were synthesized via the hydrothermal method. A total of 6.44 g of glucose (Aladdin, Shanghai, China) was dissolved into 65 mL of deionized water to form a solution at room temperature. Then, the solution was transferred into a 100 mL Teflon-lined stainless steel autoclave (Hongguan, Shanghai, China). The reaction temperature was 180 °C for 10 h. The as-obtained precipitates were collected by centrifugation [42]. Then, 5.864 g InCl_3_∙4H_2_O (Aladdin, Shanghai, China) was dissolved in 20 mL of deionized water under stirring at room temperature. Next, 200 mg as-obtained carbon templates were dispersed in the above solution, and then the suspension was stirred at room temperature for 12 h. The obtained precipitates were centrifugally washed and dried at 60 °C. The dried precipitates were calcined at 500 °C in a muffle furnace (Kejing, Fufei, Anhui, China) for 3 h to remove the carbon template.

The 50 mg as-obtained In_2_O_3_ sample was dispersed into a 25 mL ethanol/water mixture (Aladdin, Shanghai, China; the volume ratio was ethanol:water = 4:1) to form a suspension. Then, 0.3 mL tetrabutyl titanate (Aladdin, Shanghai, China) and 0.1 mL ammonia (Aladdin, Shanghai, China) were added into 25 mL of absolute ethanol (Aladdin, Shanghai, China) to form a solution, and the solution was slowly added to the In_2_O_3_ suspension. The mixture was stirred for 6 h at 80 °C in the water bath. Then, the obtained precipitates were washed by centrifugation and dried. Finally, the precipitates were calcined at 500 °C for 2 h.

The Au modified TiO_2_/In_2_O_3_ nanocomposites were obtained by the chemical reduction method [43]. First, 50 mg of the as-obtained TiO_2_/In_2_O_3_ sample was dispersed into 15 mL of deionized water to form a suspension, then 0.45 mL of 0.01 M chloroauric acid (Aladdin, Shanghai, China) and 1 mL of 0.01 M L-lysine solution (Aladdin, Shanghai, China) were added into the above suspension. After stirring for 30 min, 0.1 mL of 1 M sodium citrate solution (Aladdin, Shanghai, China) was added into the above mixture under continual stirring. The precipitate was washed with deionized water and absolute ethanol and dried at 60 °C for 12 h. Finally, the precipitates were calcined at 300 °C for 30 min in air to remove the lysine.

### 2.2. Characterization

The crystalline structures of the hollow In_2_O_3_, TiO_2_, TiO_2_/In_2_O_3_, and Au/TiO_2_/In_2_O_3_ nanocomposites were examined by an X-ray diffractometer with Cu Kα radiation (XRD, Bruker D8 Advance, λ = 0.15406 nm, Germany) operated at 40 KV. The morphology was investigated by a field-emission scanning electronic microscope (FESEM, FEI Company, Hillsboro, OR, USA, QUANTA FEG 250) and transmission electron microscope (HRTEM, Hitachi H-800, Japan). The valence state was obtained by an X-ray photoelectron spectrometer (XPS, Physical electronics, PHI 5300, Germany) with an Al Kα = 280 eV excitation source. The pore structure was obtained by Mercury Injection Apparatus (BET, Micromeritics, Autopore 9620, Norcross, GA, USA). In situ diffuse reflectance Fourier transform manufacture spectra of the samples were measured on a Fourier transform infrared spectrometer equipped (in situ DRIFTS, BRUKER OPTICS, Tensor, Germany) at a resolution of 4 cm^−1^ by accumulating 40 scans.

### 2.3. Sensor Fabrication and Sensing Measurements

An appropriate amount of deionized water or absolute ethanol was added to the powdered samples and ground into a slurry in a mortar. The slurries were evenly coated on an alumina ceramic substrate printed with gold interdigital electrodes (Beirun, Changchun, China, 15 × 8 × 0.6 mm, finger width of 0.4 mm, finger spacing of 0.2 mm, 8 pairs of electrodes). The as-fabricated sensors were aged at 60 °C for at least 24 h before the gas sensing test to ensure stability. The gas sensing performances were researched by a static testing method, as shown in Figure 1. The test system consists of a signal acquisition system, digital multimeter (Agilent 34465A, Santa Clara, CA, USA), and test chamber. The as-fabricated sensors were placed in a test chamber with a capacity of 50 L. The Agilent 34465A digital multimeter was connected to the computer to record the change curve of the resistance value of the sensors. The gas sensing test process was performed at room temperature and excited by ultraviolet light (UV-LED, λ = 365 nm, power density = 2.5 mW/cm^2^) at a distance of about 0.5 cm. In the traditional method, the front window of the test chamber is closed to ensure that the chamber is sealed, afterward, the baseline resistance of the sensor was stabilized, and the target gases or liquids were injected into the chamber through a microinjector (Collect, Zhengzhou, Henan, China). The effects of accompanying water in the formaldehyde solutions are discussed in the Appendix A. When the sensor response signal is stable, the front window of the test chamber is opened to exhaust the gas and test the recovery properties of the sensors.

The sensor response (S) was defined as:S = R_air_/R_gas_(1)
where R_air_ represents the resistance of the sensors in the air and R_gas_ represents the resistance of the sensor in targeted gas vapors.

## 3. Results

### 3.1. Structural and Morphological Characteristics

The phase and composition of the as-prepared samples were characterized by X-ray diffraction. Figure 2a shows the XRD patterns of the hollow In_2_O_3_, TiO_2_, TiO_2_/In_2_O_3_, and Au/TiO_2_/In_2_O_3_ nanocomposites. From the results, all the XRD peaks of the TiO_2_/In_2_O_3_ nanocomposites could be well-assigned to the anatase phase TiO_2_ (JCPDS card No. 21-1272) and cubic phase In_2_O_3_ (JCPDS card No. 06-0416), respectively. After Au nanoparticles were modified on the surface of the TiO_2_/In_2_O_3_ nanocomposites, the peaks of Au appeared at 38.18°, 44.39°, 64.58°, and 77.55° assigned to the (111), (200), (220), and (311) planes of face-centered cubic Au (JCPDS card No. 04-0784), respectively. In addition, the diffraction peaks of the Au/TiO_2_/In_2_O_3_ nanocomposites did not shift after decorating the Au nanoparticles, which indicates that the Au atoms were not doped into the crystal structure. In the nanocrystals, inherent strain exists due to size limitations, and this important elastic property can affect the optical and electrical properties of materials [44]. XRD analysis of the nanocrystals can confirm the crystallinity of the sample, which exhibits different peaks associated with different reflection planes. The Williamson–Hall (W–H) method is a suitable method to study various elastic properties including strain as well as calculate the average size. In the XRD data, the broadening ($\beta_{T}$) of the peaks was due to the combined effect of the crystallite size ($\beta_{D}$) and micro strain ($\beta_{\varepsilon }$).
(2)$$\beta_{T}=\beta_{D}+\beta_{\varepsilon }$$

From the Scherer equation,
(3)$$\beta_{D}=\frac{K\lambda }{D\cos\theta }$$
where $\beta_{D}$ is the full width at half maxima (FWHM) in radians, K = 0.9 is the shape factor, λ = 0.15406 nm is the wavelength of X-ray source, D is the crystallite size, and θ is the peak position in radians. The XRD peak broadening due to micro strain is given as:(4)$$\beta_{\varepsilon }=4\varepsilon \tan\theta$$
where $\beta_{\varepsilon }$ is broadening due to strain, ε is the strain, and θ is the peak position in radians. Putting Equations (3) and (4) in Equation (2):(5)$$\beta_{T}=\;\frac{K\lambda }{D\cos\theta }\;+4\varepsilon \tan\theta$$

Therefore, Equation (5) can be written as:(6)$$\beta_{T}\cos\theta =\;\varepsilon \left(4\sin\theta \right)+\frac{K\lambda }{D}$$

Equation (6) is an equation of a straight line as the uniform deformation model (UDM) equation, which considers the isotropic nature of the crystals. Figure 2b shows the UDM plots for the In_2_O_3_, TiO_2_, TiO_2_/In_2_O_3_, and Au/TiO_2_/In_2_O_3_ nanocomposites. The slope of the line provides the value of the strain, while the intercept provides the average particle size of the crystal. According to the UDM, the average particle sizes of the In_2_O_3_, TiO_2_, TiO_2_/In_2_O_3_, and Au/TiO_2_/In_2_O_3_ nanocomposites are about 23 nm, 53 nm, 29 nm, and 29 nm, respectively. The slope of the fitted lines in the UDM diagram is positive, indicating that the lattice expands to generate an intrinsic strain in the nanocrystals.

**Figure 2 materials-16-04010-f002:**
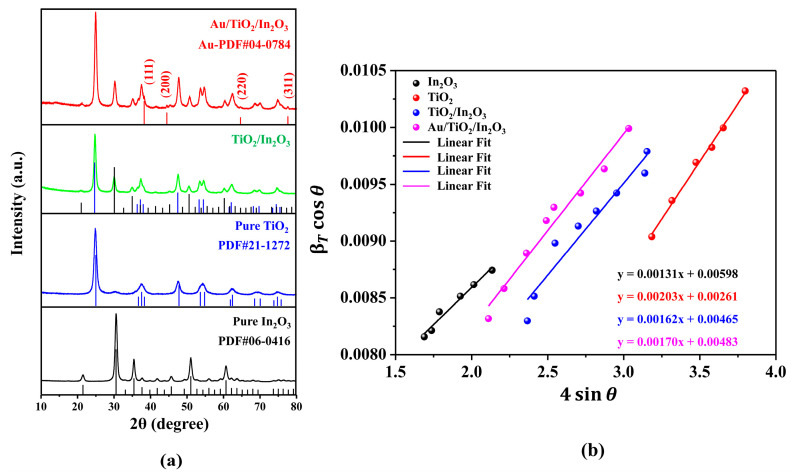
(**a**) X-ray diffraction (XRD) spectra of the hollow In_2_O_3_, TiO_2_, TiO_2_/In_2_O_3_, and Au/TiO_2_/In_2_O_3_ nanocomposites; (**b**) Uniform deformation model (UDM) plots for the In_2_O_3_, TiO_2_, TiO_2_/In_2_O_3_ and Au/TiO_2_/In_2_O_3_ nanocomposites.

Figure 3 shows the SEM images of the carbon nanospheres, hollow In_2_O_3_ nanospheres, and TiO_2_/In_2_O_3_ and Au/TiO_2_/In_2_O_3_ nanocomposites. As shown in Figure 3a, the particle size of the carbon nanospheres was relatively uniform, and the size was about 100–200 nm. Figure 3b shows the SEM image of the prepared In_2_O_3_ nanospheres. As can be seen, the In_2_O_3_ sample was composed of a large number of spherical nanoparticles, and the diameter of the In_2_O_3_ nanospheres was about 150–250 nm. The hollow structure could be observed from the broken area of some nanoparticles. As can be seen from Figure 3c, after forming the nanocomposite material with TiO_2_, the surface of the nanospheres became obviously rough, and was observed that the TiO_2_ nanoparticles were uniformly covered on the surface of the In_2_O_3_ nanospheres. The particle size was about 150–250 nm. Figure 3d shows the SEM image of the Au/TiO_2_/In_2_O_3_ nanocomposites, and as can be seen, the hollow structure of the TiO_2_/In_2_O_3_ nanocomposites was still relatively intact, and the size of the TiO_2_/In_2_O_3_ nanospheres was not significantly changed. Due to the limitation of magnification, the existence of Au nanoparticles could not be observed. Therefore, the information of the Au nanoparticles and hollow structure of the Au/TiO_2_/In_2_O_3_ nanocomposites can be further observed by high resolution TEM.

The hollow structure of the Au modified TiO_2_/In_2_O_3_ nanocomposites could be clearly observed in the TEM images, as shown in Figure 4. As it can be seen in Figure 4a,b, the particle size of the Au/TiO_2_/In_2_O_3_ nanocomposites was 150–250 nm, and it could be observed that the Au nanoparticles were evenly dispersed on the surface of the nanospheres, and the size was about 10–15 nm. From Figure 4c, the wall thicknesses of the In_2_O_3_ shell and TiO_2_ shell were about 10 nm and 15 nm, respectively. The lattice fringes could be observed with an interplanar distance of 0.294 nm and 0.353 nm, which corresponded to the In_2_O_3_ (111) and TiO_2_ (101) crystal planes, respectively. Figure 4d shows the typical HRTEM images at the interface of the Au and TiO_2_ nanoparticles. The lattice fringes of Au and TiO_2_ interlace at the interface, which may be the effect of the overlapping of different lattice fringes of Au nanoparticles and TiO_2_ nanoparticles at the interface. The interplanar distance of Au was calculated to be 0.234 nm, corresponding to the Au (111) crystal planes. Figure 4j shows the energy dispersive spectrometer (EDS) mapping of the Au modified TiO_2_/In_2_O_3_ nanocomposites. It can be seen from the energy spectrum that the sample was composed of O, In, Ti, and Au elements, which indicates that the sample is a ternary composite material. The Au element was uniformly distributed across the whole sample. The weight percentages of the Ti, In, O, and Au elements were tested as 24.92 wt%, 48.10 wt%, 25.21 wt%, and 1.77 wt%, respectively, which was close to the theoretical additions (Ti: 24.75 wt%, In: 48.54 wt%, O: 25.00 wt%, and Au: 1.71 wt%).

XPS analysis was used to characterize the element composition and valence state information of the Au/TiO_2_/In_2_O_3_ nanocomposites, as shown in Figure 5. The characteristic peak of carbon can be used as a reference (C 1s 284.6 eV) for calibration [45]. As can be seen in Figure 5a, the characteristic peaks of elements In, Ti, Au, and O were observed in the full survey scan spectrum of the sample. The weight percentages of Ti 2p, In 3d, O 1s, and Au 4f elements were calculated as 23.41 wt%, 47.57 wt%, 27.20 wt%, and 1.82 wt%, respectively, which is basically consistent with the EDS results. Figure 5b presents the high-resolution spectrum of Ti 2p. The characteristic peaks at 458.5 eV and 464.2 eV were attributed to Ti 2p_3/2_ and Ti 2p_1/2_, respectively, indicating that the valence state of the Ti element is Ti^4+^ [46]. Figure 5c shows the XPS spectrum of In 3d. It can be seen that there are two characteristic peaks at the binding energy of 444.0 eV and 451.6 eV, which corresponded to the In 3d_5/2_ and In 3d_3/2_ electron orbitals, respectively [47]. The results showed that the In element exists in the form of In^3+^. Figure 5d shows the XPS spectrum of Au 4f. There are two characteristic peaks at the binding energy of about 83.1 eV and 86.7 eV, which corresponded to the electron orbitals of Au 4f_7/2_ and Au 4f_5/2_, respectively [48]. In addition, the O 1s spectrum of the TiO_2_/In_2_O_3_ and Au/TiO_2_/In_2_O_3_ samples were analyzed as shown in Figure 5e–f. The O1s spectrum was fitted and analyzed by the Gaussian fitting method, and three characteristic peaks were obtained, which corresponded to the lattice oxygen (O_L_), oxygen vacancies (O_v_), and chemisorbed oxygen (O_C_) [49,50,51], respectively. As shown in Figure 5f, the peaks of O_L_, O_V_, and O_C_ were located at about 529.32 eV, 530.10 eV, and 531.63 eV, respectively. The content of different oxygen ions before and after Au modification was analyzed, and the results are shown in Table 1. Compared with the TiO_2_/In_2_O_3_ samples, the oxygen vacancy content ratio of the Au/TiO_2_/In_2_O_3_ increased from 20.5% to 36.0%, and the chemisorbed oxygen content ratio increased from 11.5% to 23.7%. Chemisorbed oxygen will participate in the subsequent gas sensing response, therefore, the increase in chemisorbed oxygen content on the sample surface is conducive to the improvement in sensing performance. When TiO_2_ is irradiated by UV light, the electrons in the valence band are excited to the conduction band, where electrons and holes migrate to the surface of TiO_2_, forming electron–hole pairs on the surface; electrons react with Ti^4+^, and holes react with surface bridging oxygen ions to form Ti^3+^ and oxygen vacancy, respectively. However, the loading of Au nanoparticles on the surface of TiO_2_ can further promote the generation of electron–hole pairs, thus promoting the formation of oxygen vacancy. Therefore, the relative content of oxygen vacancy increases in the O 1 s spectra of the Au/TiO_2_/In_2_O_3_ nanocomposites.

In order to further characterize the porous structures of the Au/TiO_2_/In_2_O_3_ nanocomposites, nitrogen adsorption–desorption tests were conducted, and the results are shown in Figure 6. The curves show typical IV-type isotherm characteristics with H3-type hysteresis loops, which indicates that the sample had a mesoporous structure [52]. According to the pore size distribution curves, the pore size was mainly distributed around 7.5 nm. The specific surface area of the sample was 49.73 m^2^/g. Mesoporous structures can promote the diffusion, adsorption and desorption of target gas molecules, and it also provides more chemical active sites for the reaction of gas molecules with adsorbed oxygen ions on the surface, which plays an important role in improving the gas sensing properties.

### 3.2. Electrical Characterization and Gas-Sensing Property

Figure 7a shows the I–V polarization curves of sensors based on the TiO_2_/In_2_O_3_ nanocomposites and the Au/TiO_2_/In_2_O_3_ nanocomposites under the UV activation. At room temperature, the I–V characteristic curves of these two sensors were linear without UV excitation, which indicates the contact type between the materials and the gold interdigital electrodes. Under UV activation, the I–V polarization curves of the two sensors showed a slight nonlinear relationship between the applied voltage and the measured current. Due to the presence of nanoheterojunctions in the samples, photogenerated electrons were generated and transferred under UV excitation [53]. This process increases the energy barrier at the nanoheterojunctions, which illustrates that UV activation enhances the nanoheterojunction properties. Figure 7b shows the photo–response curves of hollow In_2_O_3_ and TiO_2_ nanospheres at room temperature. When the UV-LED was turned on, the resistance of the TiO_2_ nanosphere-based sensor decreased rapidly, and after the UV-LED was turned off, the resistance quickly returned to the initial resistance, and the resistance change rate was about 52%. In contrast, the respond and recovery speeds of the In_2_O_3_ nanosphere-based sensor were slow for the UV-LED. The photo–response curves of the sensor based on TiO_2_/In_2_O_3_ and Au/TiO_2_/In_2_O_3_ nanocomposites to UV activation at room temperature is shown in Figure 7c. The baseline resistance of the TiO_2_/In_2_O_3_ and Au/TiO_2_/In_2_O_3_ nanocomposite-based sensors was about 7 MΩ and 33 MΩ, respectively, which was much lower compared to the baseline resistance of the TiO_2_ nanospheres (~630 MΩ). The TiO_2_/In_2_O_3_ and Au/TiO_2_/In_2_O_3_ nanocomposite-based sensor also had a fast response to UV-LED, and the response signal was very stable. When UV light irradiates to the surface of the sensing material, a large number of photogenerated electron–hole pairs will be generated, and for the Fermi level to be in equilibrium, more electrons tend to be transferred from the TiO_2_ to the Au nanoparticle side, so the Au nanoparticles on the surface act as electron acceptors, which promotes the effective separation of electrons and holes at the interface, thus promoting the generation of more photogenerated charge carriers. In this process, the resistance of the sensor changes greatly. Therefore, Au modification improves the UV activation of the Au/TiO_2_/In_2_O_3_ nanocomposites.

Figure 8a shows the response curve of the sensor based on pure In_2_O_3_, TiO_2_, TiO_2_/In_2_O_3_, and Au/TiO_2_/In_2_O_3_ nanocomposites to 1 ppm formaldehyde under UV activation at room temperature. When formaldehyde gas is introduced into the test environment, the response signals of the In_2_O_3_ and TiO_2_-based sensors were significantly less than those of the nanocomposites, while the response curves of the TiO_2_/In_2_O_3_ and Au/TiO_2_/In_2_O_3_-based sensors increased rapidly and reached a stable state. The response of the In_2_O_3_, TiO_2_, TiO_2_/In_2_O_3_, and Au/TiO_2_/In_2_O_3_ nanocomposite-based sensors were about 1.6, 2.1, 3.8, and 5.6, respectively. It can be observed that the response/recovery time of a pure In_2_O_3_-based sensor was the longest (136/233 s). In contrast, the response/recovery times of both the TiO_2_/In_2_O_3_ and Au/TiO_2_/In_2_O_3_-based sensors were significantly improved under the same test conditions, which were about 28/50 s and 18/42 s, respectively. This indicates that the addition of Au nanoparticles can effectively improve the adsorption and desorption rate of gas molecules on the surface of the sensing materials. Figure 8b shows the repeatability of response of the In_2_O_3_, TiO_2_, TiO_2_/In_2_O_3_ and Au/TiO_2_/In_2_O_3_-based sensors to 1 ppm formaldehyde for five consecutive cycles. It can be seen that in the recovery process, the resistance of the sensor can be completely restored to the initial state. The response value deviation of the Au/TiO_2_/In_2_O_3_-based sensor was less than 2%, which indicates that the Au/TiO_2_/In_2_O_3_-based sensor had good signal repeatability and stability. In can be seen from Figure 8c that the response curves of these four sensors to formaldehyde had concentrations ranging from 30 ppb to 10 ppm at room temperature. As can be seen, with the increase in the formaldehyde concentration, the response of these four sensors also increased correspondingly. The detection limit of the Au/TiO_2_/In_2_O_3_-based sensor was 60 ppb. Figure 8d shows the relationship between the response of the sensors and the formaldehyde concentrations. As can be seen, the response of the Au/TiO_2_/In_2_O_3_-based sensor was much higher than the other three sensors. These four sensors showed a good linear relationship between the response and formaldehyde concentrations. The slope of the calibration plot of the Au/TiO_2_/In_2_O_3_-based sensor was 2.64, which was significantly higher than the other sensors.

Figure 9a shows the response of the In_2_O_3_, TiO_2_, TiO_2_/In_2_O_3_, and Au/TiO_2_/In_2_O_3_-based sensors to 1 ppm of various gases under UV activation at room temperature. We compared the response value of the Au/TiO_2_/In_2_O_3_-based sensor to 1 ppm formaldehyde gas with the response of 1 ppm of acetone (S = 3.1), ammonia (S = 2.6), methanol (S = 3.7), ethanol (S = 3.1), toluene (S = 2.7), and benzene (S = 2.4); the anti-interference ability (Selectivity coefficient: S = S_HCHO_/S_interference_) of the Au/TiO_2_/In_2_O_3_-based sensor was improved through a comparison with these common interference gases. This indicates that the selectivity of Au/TiO_2_/In_2_O_3_-based sensors is improved by Au modification. In order to research the effect of ambient humidity on the gas sensing performance of the sensors, a different humidity is introduced as a comparison condition during the test, and the results are shown in Figure 9b. As can be seen, the response values of these four sensors showed a gradual decline trend with the increase in humidity. When the humidity exceeded 75%, the pure In_2_O_3_ and TiO_2_ sensors had no response signals. When the humidity was 85%, the response value of the Au/TiO_2_/In_2_O_3_-based sensor maintained 55% of the original value. This indicates that the moisture resistance of the sensor is improved after the modified Au nanoparticles. Long-term stability is considered as one of the important indices to measure the performance of sensors. Figure 9c shows the stability of the Au/TiO_2_/In_2_O_3_-based sensor to 1 ppm formaldehyde at room temperature for 90 days. It can be observed that the response curve of the sensor had not changed significantly. It was calculated that the variation range of the response value was less than 3.5%. This demonstrates that the Au/TiO_2_/In_2_O_3_-based sensor has great long-term stability for formaldehyde detection at room temperature. Table 2 summarizes the comparison of the gas sensing properties of the Au/TiO_2_/In_2_O_3_ nanocomposite-based sensor and other sensing materials reported in the literature to formaldehyde [40,43,54,55,56,57]. It can be seen that the Au modified TiO_2_/In_2_O_3_-based sensor had a low detection limit, and its response value, response/recovery time, selectivity, and stability were excellent. These results indicate that the Au/TiO_2_/In_2_O_3_ sample has good application prospects as sensing materials for the detection of formaldehyde.

Figure 10 shows the sensing mechanism of the Au modified TiO_2_/In_2_O_3_ nanocomposites to formaldehyde under UV activation at room temperature. When the sensor is in the air, the Au/TiO_2_/In_2_O_3_ is activated to produce photogenerated electron–hole pairs under the excitation of UV light. O_2_, as a strong oxidizing gas, can be quickly adsorbed on the surface of the material under UV activation and react with photogenerated electrons to form chemisorbed oxygen ions (O_2_^−^, O^2−^, O^−^). In this process, the resistance of the sensor increases. When the operating temperature is less than 100 °C, the chemisorbed oxygen ion type is O_2_^−^. The photogenerated holes are then combined with OH^−^ to form the neutral OH∙ with higher oxidation properties. The equation for the reaction is as follows:hυ → h^+^ + e^−^(7)
O_2_ + e^−^ → O_2_^−^ (hυ)(8)
h^+^ + OH^−^ → OH∙(9)

When the sensor is in contact with formaldehyde gas, the formaldehyde gas molecules will react with the pre-absorbed chemisorbed oxygen ions O_2_^−^, and the released electrons in the reaction will return to the conduction band of the oxide; during this process, the resistance of the sensor decreases [58]. The chemical reactions on the surface of the sensing materials during this process were analyzed by in situ DRIFTS spectra detection.

In situ DRIFTS spectra detection technology plays a very important role in the real-time detection of the gas adsorption and reaction process on the surface of sensors, which contributes to speculating on the reaction mechanism. Figure 11a–c shows the in situ DRIFTS spectra of the TiO_2_, In_2_O_3_, and Au/TiO_2_/In_2_O_3_ nanocomposites exposed to air and formaldehyde at room temperature under UV activation. As can be seen, the vibration absorption peaks of formaldehyde molecules can be observed at about 1060 cm^−1^ and 1150 cm^−1^. It indicates that some formaldehyde is adsorbed on the surface sensing materials in a molecular state at room temperature [59]. The dissociation of water from the air on a TiO_2_ surface forms two distinctive hydroxyl groups: one OH- group bridges two Ti^4+^ (3741 cm^−1^) and the other forms a terminal Ti^4+^–OH- group (3670 cm^−1^), which correspond to the observed bands at 3650 and 3742 cm^−1^, respectively [60,61]. From Figure 11c, the band at 2936 cm^−1^ corresponded to the characteristic absorption peak of dioxymethylene (H_2_COO), and the intensity increased significantly with the reaction, which indicates that the formaldehyde molecules adsorbed are first oxidized to dioxymethylene, so one possible involved redox reaction can be proposed as:HCHO (ads) + O_2_^−^ (ads) → H_2_COO (ads) + e^−^(10)

Compared with the absorption peaks of TiO_2_ and In_2_O_3_, the relative strength of the absorption peaks of the nanocomposites was higher, which indicates that the Au/TiO_2_/In_2_O_3_ nanoheterojunctions greatly promoted the chemical reaction of the oxidation of formaldehyde to dioxymethylene. The bands at 2835 cm^−1^, 1591 cm^−1^, 1460 cm^−1^, and 1320 cm^−1^ were the absorption peaks of formate (HCOO^−^), and their intensities increased with the extension of reaction time. Among them, 2835 cm^−1^ is the C–H stretching vibration absorption peak of formate; 1591 cm^−1^ is the asymmetric stretching vibration absorption peak of COO; 1320 cm^−1^ is the symmetric stretching vibration absorption peak of COO; 1460 cm^−1^ is the C–H bending vibration absorption peak of formate [62]. The presence of these peaks indicates that dioxymethylene is further oxidized by chemisorbed oxygen ions to form formate, as shown in the equation as follows:H_2_COO (ads) + O_2_^−^ → HCOO^−^ (ads) + OH^−^ (ads)(11)

The bands at 2328 cm^−1^ and 2358 cm^−1^ are the characteristic absorption peaks of CO_2_, and the intensity of the peaks is relatively strong, indicating that the generated formate was further oxidized, as shown in in equation as follow:HCOO^−^ (ads) + OH^−^ (ads) →CO_2_ + H_2_O + 2e^−^(12)

In situ DRIFTS spectra were used to analyze the changes of adsorbates on the surface of sensing materials during the sensing process. It was concluded that formaldehyde was first oxidized to H_2_COO, then oxidized to HCOO^−^, and finally to CO_2_. The reaction process was more intense on the surface of the Au/TiO_2_/In_2_O_3_ nanocomposites, and UV light promoted the oxidation of formaldehyde to CO_2_. Figure 11d shows the in situ DRIFTS spectra of the Au/TiO_2_/In_2_O_3_ nanocomposites without UV activation. The bands at 1062 cm^−1^ and 1142 cm^−1^ were due to the vibration absorption peaks of the formaldehyde molecules. This indicates that formaldehyde molecules can be adsorbed on the surface of sensing materials without UV light. The presence of bands at 2836 cm^−1^, 1540 cm^−1^, and 1440 cm^−1^ indicates that a small amount of formaldehyde was oxidized to H_2_COO and HCOO^−^, which also illustrates that a large number of formaldehyde molecules adsorbed on the surface of the Au/TiO_2_/In_2_O_3_ nanocomposites without significant dissociation. No characteristic absorption peak of CO_2_ was observed in the spectra, which indicates that the HCOO^−^ on the surface cannot be further oxidized to CO_2_ without UV activation. The results show that UV activation can significantly enhance the catalytic conversion of formaldehyde gas.

There are two reasons for the obvious improvement in the gas sensing of Au modified TiO_2_/In_2_O_3_ nanocomposites. On one hand, due to the different positions of the Fermi levels of In_2_O_3_ and TiO_2_, these two oxides contact and form a nanoheterojunction at the interface, so electrons will transfer from the conduction band of In_2_O_3_ to TiO_2_ through the interface and accumulate [63]. This can effectively separate the photogenerated electron–hole pair at the interface and increase the lifetime of the photogenerated electron–hole pair. With the increase in the electron content in the TiO_2_ nanoparticles, more formaldehyde molecules can participate in the REDOX reaction, which further increases the range of conductivity of the sensing materials, which improves the gas sensing properties of the Au/TiO_2_/In_2_O_3_ nanocomposites [64]. In addition, the relatively thin shell thickness also facilitates UV light to penetrate into the interior of the hollow materials. On the other hand, the modification of noble metals also plays an important role in improving the gas sensing performance of Au/TiO_2_/In_2_O_3_. First, due to the “spillover effect” of Au, Au nanoparticles modified on TiO_2_/In_2_O_3_ nanospheres can be used as a chemical sensitizer [65]. The catalytic capacity of noble metal nanoparticles for oxygen decomposition is much higher than that of In_2_O_3_ and TiO_2_, and the reactive oxygen species obtained after catalytic decomposition will overflow to the surface of the sensing material, which increases the width of the electron depletion layer of the sensing material and further increases the resistance of the sensing material in the air [66]. In addition, noble metal modification can significantly increase the proportion of active oxygen in Au/TiO_2_/In_2_O_3_, which promotes the reaction between formaldehyde and O_2_^−^ and improves the sensing of the sensor, which is consistent with the results of XPS.

## 4. Conclusions

Au modified TiO_2_/In_2_O_3_ hollow nanospheres were prepared via a facile hydrolysis method with the assistance of a template. The results of the TEM, EDS, and XPS showed that Au nanoparticles were successfully modified on the surface of the TiO_2_/In_2_O_3_ nanocomposites, and the Au/TiO_2_/In_2_O_3_ nanocomposites maintained a double-layer hollow nanosphere structure. The size of the Au/TiO_2_/In_2_O_3_ nanospheres was about 150–250 nm. The thickness of the outer TiO_2_ shell and inner In_2_O_3_ wall was about 15 nm and 10 nm, respectively. The size of the Au nanoparticles was about 10–15 nm. Compared with the pure In_2_O_3_ and TiO_2_-based sensor, the Au modified TiO_2_/In_2_O_3_ nanocomposite-based sensor had better gas sensing properties to formaldehyde. The response of the Au/TiO_2_/In_2_O_3_ nanocomposite-based sensor to 1 ppm formaldehyde was about 5.6, and the detection limit was 60 ppb. The response and recovery times of the Au/TiO_2_/In_2_O_3_ nanocomposite-based sensor were 18 s and 42 s, respectively, which was faster than that of the pure In_2_O_3_, TiO_2_, and TiO_2_/In_2_O_3_ nanocomposites. The results of the in situ DRIFTS show that the nanoheterojunctions promoted the catalytic conversion of formaldehyde on the surface of the Au/TiO_2_/In_2_O_3_ nanocomposites under UV activation. The improvement in the gas sensing properties of the Au/TiO_2_/In_2_O_3_ nanocomposites is mainly due to the electronic and chemical sensitization of the Au nanoparticles.

## Figures and Tables

**Figure 1 materials-16-04010-f001:**
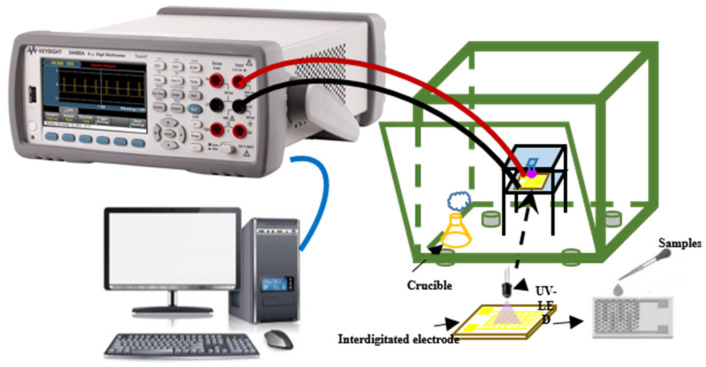
The schematic diagram of the sensing measurement apparatus.

**Figure 3 materials-16-04010-f003:**
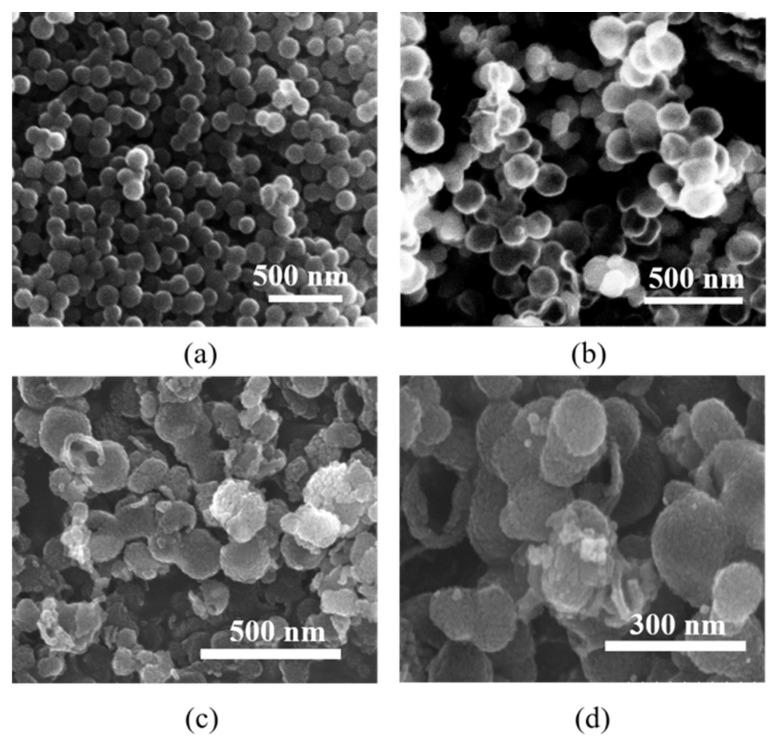
Scanning electron microscope (SEM) images of (**a**) carbon nanospheres; (**b**) hollow In_2_O_3_ nanospheres; (**c**) TiO_2_/In_2_O_3_ nanocomposites, and (**d**) Au/TiO_2_/In_2_O_3_ nanocomposites.

**Figure 4 materials-16-04010-f004:**
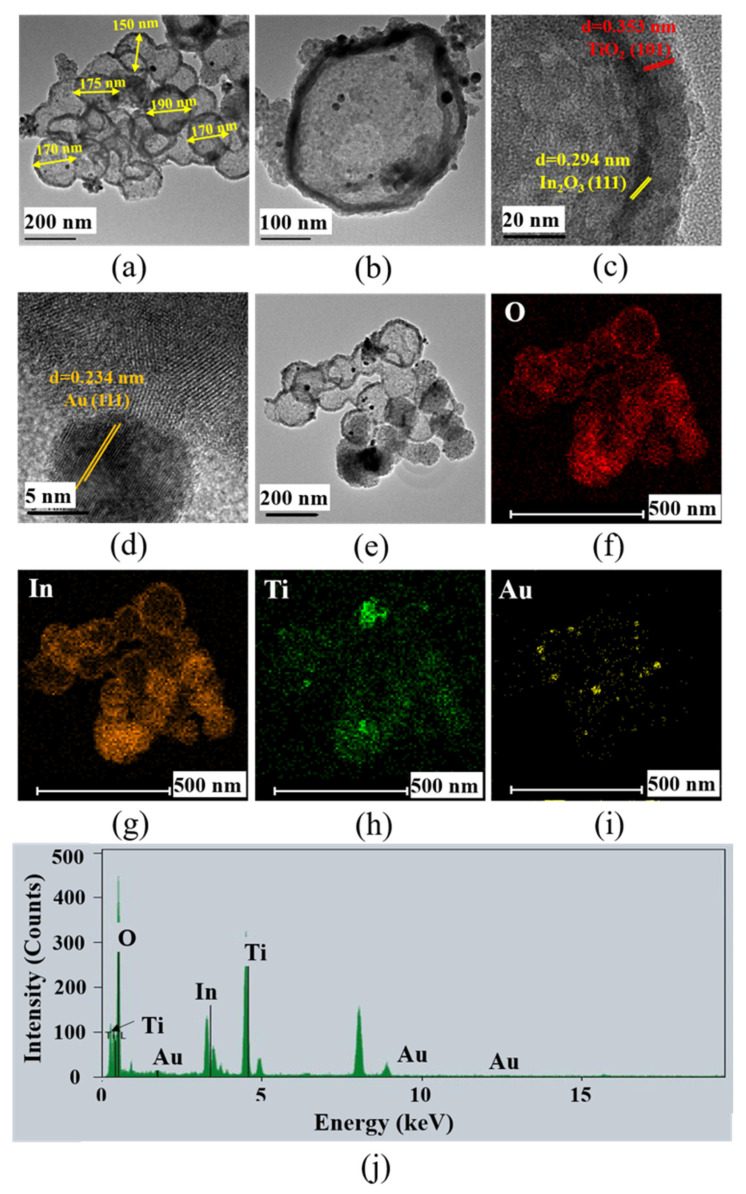
(**a**–**e**) Transmission electron microscope (TEM) and high resolution-TEM images of the Au/TiO_2_/In_2_O_3_ nanocomposites; (**f**–**j**) Energy dispersive spectrometer (EDS) mapping of the Au/TiO_2_/In_2_O_3_ nanocomposites for the overlay, O element, Ti element, In element, and Au element.

**Figure 5 materials-16-04010-f005:**
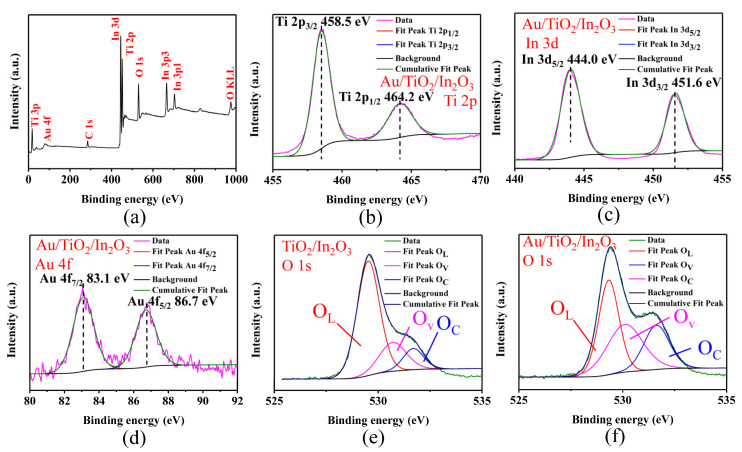
X-ray photoelectron spectroscopy (XPS) spectra of: (**a**) full survey scan spectrum; (**b**) Ti 2p spectrum of Au/TiO_2_/In_2_O_3_; (**c**) In 3d spectrum of Au/TiO_2_/In_2_O_3_; (**d**) Au 4f spectrum of Au/TiO_2_/In_2_O_3_; (**e**) O 1s spectrum of TiO_2_/In_2_O_3_; (**f**) O 1s spectrum of Au/TiO_2_/In_2_O_3_.

**Figure 6 materials-16-04010-f006:**
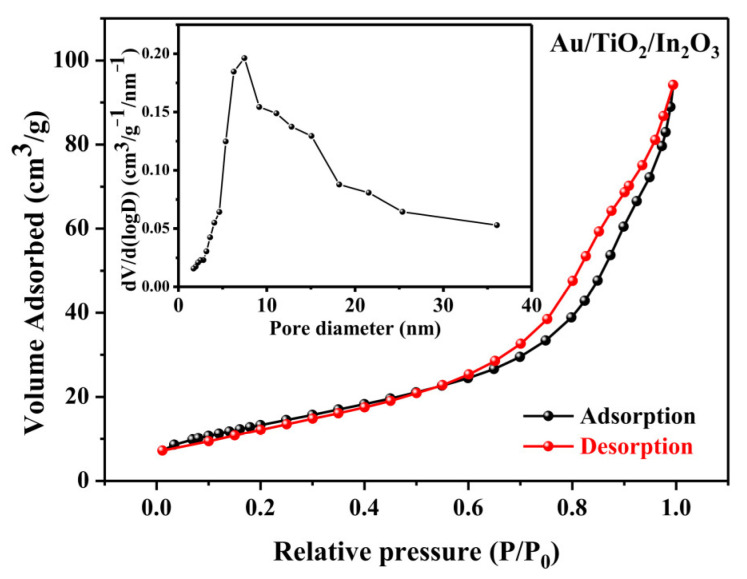
N_2_ adsorption–desorption isotherms and the pore size distribution plots (inset) of the Au/TiO_2_/In_2_O_3_ nanocomposites.

**Figure 7 materials-16-04010-f007:**
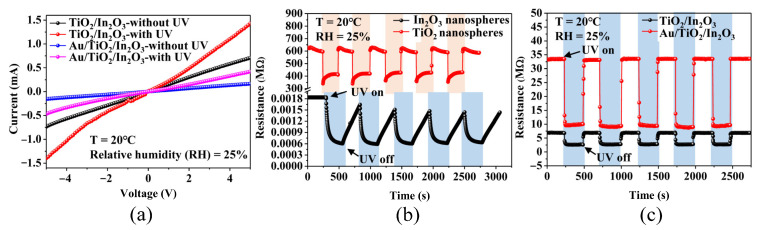
(**a**) I–V polarization curves of the TiO_2_/In_2_O_3_ and Au/TiO_2_/In_2_O_3_ nanocomposite-based sensors at room temperature; (**b**) photo–response curves of In_2_O_3_ and TiO_2_ nanosphere-based sensors to UV-LED at room temperature; (**c**) photo–response curves of TiO_2_/In_2_O_3_ and Au/TiO_2_/In_2_O_3_-based sensors to UV-LED at room temperature.

**Figure 8 materials-16-04010-f008:**
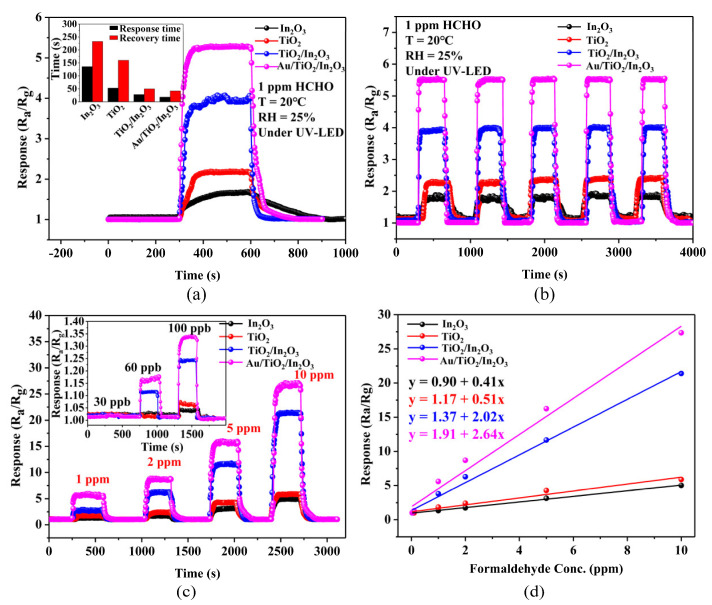
(**a**) Response curves of the In_2_O_3_, TiO_2_, TiO_2_/In_2_O_3_, and Au/TiO_2_/In_2_O_3_-based sensors to 1 ppm HCHO under UV at room temperature; (**b**) repeatability of response of the In_2_O_3_, TiO_2_, TiO_2_/In_2_O_3_, and Au/TiO_2_/In_2_O_3_-based sensors to 1 ppm HCHO; (**c**) response curves of the In_2_O_3_, TiO_2_, TiO_2_/In_2_O_3_, and Au/TiO_2_/In_2_O_3_-based sensors to 0.03–10 ppm HCHO; (**d**) correlation of the response and HCHO concentrations of the sensors (where R_a_ represents the resistance of the sensors in the air and R_g_ represents the resistance of the sensor in targeted gas vapors).

**Figure 9 materials-16-04010-f009:**
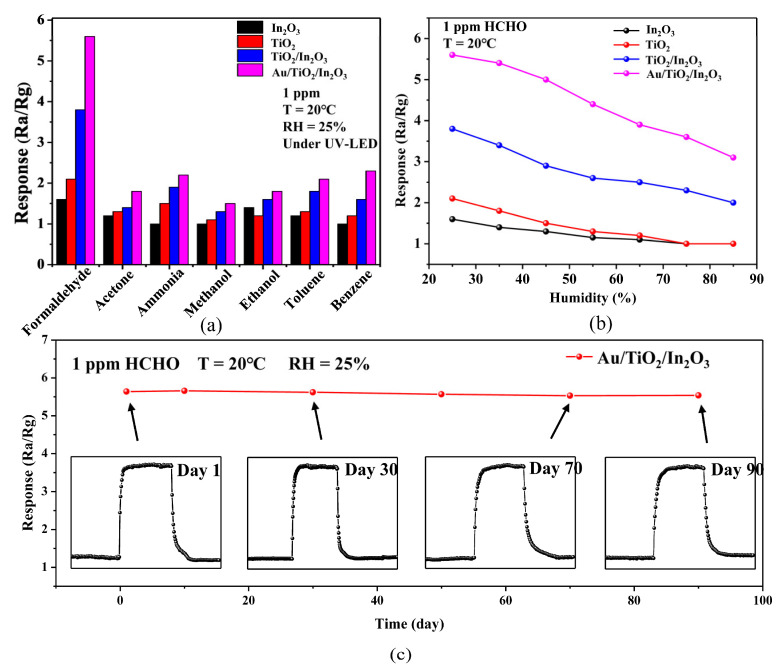
(**a**) Response of the In_2_O_3_, TiO_2_, TiO_2_/In_2_O_3_, and Au/TiO_2_/In_2_O_3_-based sensors to 1 ppm of various gases under UV activation at room temperature; (**b**) response of sensors to 1 ppm HCHO under various humidity; (**c**) the long-term stability of the Au/TiO_2_/In_2_O_3_ nanocomposite-based sensor for 90 days under UV activation at room temperature.

**Figure 10 materials-16-04010-f010:**
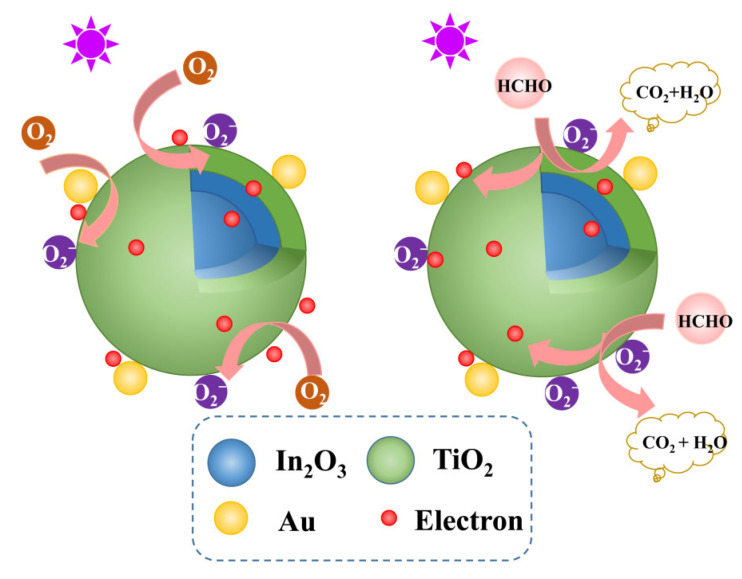
Schematic illustration of the sensing process of the Au/TiO_2_/In_2_O_3_ nanocomposite-based sensor under UV activation at room temperature.

**Figure 11 materials-16-04010-f011:**
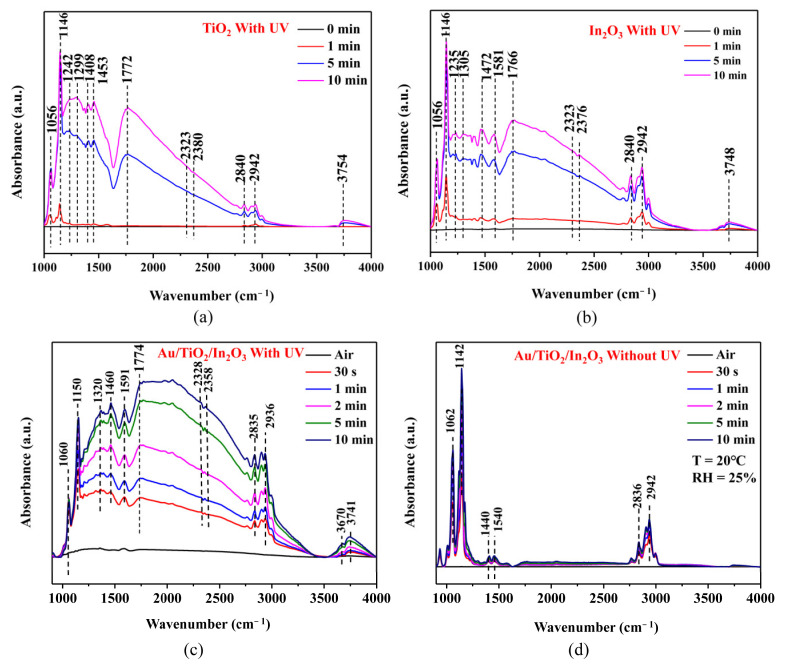
In situ DRIFTS spectra of the (**a**) TiO_2_, (**b**) In_2_O_3_, and (**c)** Au/TiO_2_/In_2_O_3_ nanocomposites exposed to air and HCHO at room temperature with UV activation; (**d**) in situ DRIFTS spectra of the Au/TiO_2_/In_2_O_3_ nanocomposites exposed to air and HCHO at room temperature without UV activation.

**Table 1 materials-16-04010-t001:** Curve-fitting results of the high-resolution X-ray photoelectron spectroscopy (XPS) spectra for the O 1s region.

Sample	O_L_		O_V_		O_C_	
	E_b_ (eV) ^1^	*r*_i_ (%) ^2^	E_b_ (eV)	*r*_i_ (%)	E_b_ (eV)	*r*_i_ (%)
TiO_2_/In_2_O_3_	529.52	68.0	530.63	20.5	531.68	11.5
Au/TiO_2_/In_2_O_3_	529.32	40.3	530.10	36.0	531.63	23.7

^1^ E_b_ (eV) represents the binding energy of peaks; ^2^
*r*_i_ (%) represents the ratio *A*_i_/∑*A*_i_ (*A*_i_ is the area of each peak).

**Table 2 materials-16-04010-t002:** Comparison of the gas sensing properties of the sensors based on different materials to formaldehyde reported in the literature.

Sensing Materials	T (°C)	Response	Conc. (ppm)	Range (ppm)	Res./Recov. (s)	Ref.
Au/In_2_O_3_	100	85.67	50	0.00142–100	28/198	[40]
Au/In_2_O_3_	200	17	100	5–100	7/135	[43]
Au/TiO_2_	RT (UV) *	8.5	5	0.1–10	36/110	[54]
Au/SnO_2_	RT (UV)	2.9	50	20–50	80/62	[55]
Au/ZnO	70	68.8	100	1–100	216/106	[56]
Au/ZnO	220	11.74	10	10–1500	38/18	[57]
Au/TiO_2_/In_2_O_3_	RT (UV)	5.6	1	0.03–10	18/42	This work

* RT (UV): Room temperature at ultraviolet activation.

## Data Availability

The data presented in this study are available on request from the corresponding author.

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
