# Peer review of "UV-Activated Au Modified TiO2/In2O3 Hollow Nanospheres for Formaldehyde Detection at Room Temperature"

_materials, 2023, doi:10.3390/ma16114010_

Round 1

Reviewer 1 Report

In this study, the authors fabricated Au-modified In2O3-TiO2 heterostructures for gas sensor applications. The performance of the fabricated sensors was improved by UV exposure, and in-situ diffuse reflectance infrared Fourier transform (FT-IR) spectroscopy was performed during the chemi-resistive gas sensor measurements. Although DRIFT measurements are of great interest, the novelty of the sensing layer is lacking. In particular, the interpretation of the results, comparison with the literature, and the motivation should be improved. Both qualitative and quantitative major revisions are required. The authors need to meet the following requirements:

1-    Why did the authors choose the In2O3/TiO2 heterostructure for gas sensor applications? In the introduction part, TiO2 and In2O3 were separately mentioned, but what is the importance of forming the heterostructure (particularly In2O3/TiO2) for gas sensing applications, especially against formaldehyde?

2-      In this study, the authors performed three different approaches to improve the sensor performance: heterostructure formation, spill-over effect, and UV exposure. Each individually is used to improve sensing performance of MOXs materials However, it is not clear from the introduction why the authors combine all of them. From the reader's point of view, it appears that the authors tried every known method to improve the sensor performance, but the scientific basis for combining these methods was not explained.

3-   At the end of the fourth paragraph, it would be helpful for the authors to provide specific examples of the spill-over effect of Au against formaldehyde, as this would illustrate the relevance of this effect to the gas sensing application studied.

4-     In section 2.1, it is not clear how the authors obtained the fabrication recipes for In2O3 nanospheres, In2O3/TiO2 heterostructures, and Au chemical reduction. Relevant literature should be added to explain the methods used to fabricate these structures.

5-    In section 2.3, more information should be provided about the sensor setup. Specifically:

a.       How did the authors obtain the " alumina ceramic substrate printed with gold interdigital electrodes"?

b.       How were the target gases generated?

c.       Which gas was introduced to the system to obtain a baseline (N2, dry air, etc.)?

d.       How was the recovery operation performed?

6- In line 147, the authors mentioned that "...the target gases or liquids were injected into the chamber through a microinjector." What are the "liquids" that were used in the experiments? And how they were used?

7.       In section 3.1, the authors investigated the crystal structure of the fabricated samples using the XRD method. As it is known, strains directly affect the electrical properties of materials. Therefore, calculating the strain values using the Williamson-Hall approach would provide important information for explaining the sensor results.

8.       The production steps involved first In2O3, then TiO2, and finally Au. In this case, the final structure produced should be Au/TiO2/In2O3. However, the authors specified Au/In2O3/TiO2 in the title and text. It is not clear why the order is different in the title and text.

9.       In line 183, the authors mentioned that "...it can be observed that TiO2 nanoparticles are uniformly covered on the surface of In2O3 nanospheres". Thus, only TiO2 reacts with the target gases and In2O3 behaves like conduction paths. This raises concerns about the detection mechanism described in the article. Therefore, the detection mechanism should be explained based on the fabricated structure to clarify this issue.

10.   Calculating the Ti-In-O-Au ratios from XPS analysis and comparing them with EDS results would increase the quality of the study.

11.   Au contacts generally form Schottky Contacts with semiconductor materials such as MOXs, TMDs, etc. (https://doi.org/10.1063/1.4840317). However, in this study, it was observed that the authors obtained ohmic contact on Au/In2O3/TiO2, as seen in Fig. 7a. It would be beneficial to explain this observation in detail.

12.    In part 3.2, it was observed that Au modification improves the UV activation of the heterostructures, as seen in Fig. 7c. The reason for this effect of Au should be explained in more detail.

13.   The sensitivity of the heterostructure to Relative humidity is observed to be twice that of the individual responses of TiO2 and In2O3, as seen in Fig. 9b. However, this result should be explained in more detail in the text.

14.   References 53 and 55 appear to be incorrect and require correction.

15.   The DRIFT measurements conducted by the authors to understand the sensing mechanism are a significant contribution to the study. However, it is noticed that the importance of these measurements was not emphasized in the Introduction, Abstract, and Conclusion sections of the article. Moreover, there is no explanation about what DRIFT stands for. Therefore, it is recommended to highlight the importance of these measurements throughout the article and emphasize the results. Also DRIFT can be addedto the list of keywords.

16.   Though DRIFT was used to investigate the sensing mechanism, comparing the DRIFT results of TiO2, In2O3, and the heterostructure was missing. Since heterostructure is one of the main motivations, DRIFT measurements of TiO2, In2O3 and heterostructure should be investigated and discussed in sensing mechanism.

17.   There is a significant disconnect between experimental studies. The authors have carried out important experimental studies such as XRD, XPS, and DRIFT that could contribute greatly to the detection mechanism, but did not put any emphasis on these results when describing the sensing  mechanism (last two paragraphs of Chapter 3). Strong emphasis should be placed on these studies when explaining the detection mechanism.

Author Response

Dear Reviewer,

We appreciate your comments and recommendations. Your comments and suggestions are very helpful for improving our research works and the paper quality. We carefully went through all the comments given by the reviewers, and we have made necessary modifications/corrections point-by-point according to the suggestions and comments. The changes introduced in the revised manuscript are highlighted in yellow. Please see the attachment.

Author Response

(The authors gave the same response as above.)

Reviewer 3 Report

This work reports formaldehyde sensor based on UV-activated Au-In2O3-TiO2 nanocomposite. The nanocomposite performance is higher than its constituents. The work has been done systematically with proper control.

I would recommend this work to be published after minor revision according to these points:

1. Abstract: Please mention the unit of 5.6 and put the number in perspective i.e. compare it with that of the constituents.

2. Methods:

- Explain in more detail how the carbon nanosphere were synthesized or add relevant reference if any.

- How was the interdigitated electrodes were obtained?

- How was the formaldehyde obtained? How was the concentration controlled?

- How was the humidity generated, controlled and monitored?

3. Figure 4: The annotations are very difficult to read due to small font size and poor resolution.

4. Figure 5 - XPS: Linear baseline is known to be less accurate. Please consider a more accurate baseline correction for example: Shirley type correction.

5. Table 1: Oxygen vacancy increases significantly after Au introduction. Could the authors explain the reason?

6. Figure 7 b and c: Add notation where the UV is turn on and off.

7. The response-recovery time of In2O3-TiO2 is faster than TiO2. Could the author explain the reason? Is it related to the different nanostructure e.g. pore size between those samples?

8. Figure 9: In terms of percentage, In2O3-TiO2 and TiO2 response also dropped to around 50% of the response in dry condition. How could the author claimed that Au improve the humidity resistance?

9. Long term test. How was the sensor stored in-between the data point? Was it under constant UV exposure all the 90 days long?

Author Response

(The authors gave the same response as above.)

Round 2

Reviewer 1 Report

Authors have met all requirements. The paper were quite improved. Thus, it can be published.

Reviewer 2 Report

The authors have modified the manuscript according to my suggestions and those other reviewers’. In my opinion, in this new form, the manuscript can be published.